# Astaxanthin Carotenoid Modulates Oxidative Stress in Adipose-Derived Stromal Cells Isolated from Equine Metabolic Syndrome Affected Horses by Targeting Mitochondrial Biogenesis

**DOI:** 10.3390/biom12081039

**Published:** 2022-07-27

**Authors:** Malwina Mularczyk, Nabila Bourebaba, Krzysztof Marycz, Lynda Bourebaba

**Affiliations:** 1Department of Experimental Biology, Faculty of Biology and Animal Science, Wrocław University of Environmental and Life Sciences, Norwida 27B, 50-375 Wrocław, Poland; nabila.bourebaba@upwr.edu.pl (N.B.); k.marycz@mimt.com.pl (K.M.); 2International Institute of Translational Medicine, Jesionowa 11, Malin, 55-114 Wisznia Mała, Poland

**Keywords:** astaxanthin, antioxidant, equine metabolic syndrome, ASCs, mitochondria, OXPHOS

## Abstract

Astaxanthin is gaining recognition as a natural bioactive component. This study aimed to test whether astaxanthin could protect adipose-derived stromal stem cells (ASCs) from apoptosis, mitochondrial dysfunction and oxidative stress. *Phaffia rhodozyma* was used to extract astaxanthin, whose biocompatibility was tested after 24, 48 and 72 h of incubation with the cells; no harmful impact was found. ASCs were treated with optimal concentrations of astaxanthin. Several parameters were examined: cell viability, apoptosis, reactive oxygen levels, mitochondrial dynamics and metabolism, superoxide dismutase activity, and astaxanthin’s antioxidant capacity. A RT PCR analysis was performed after each test. The astaxanthin treatment significantly reduced apoptosis by modifying the normalized caspase activity of pro-apoptotic pathways (p21, p53, and Bax). Furthermore, by regulating the expression of related master factors SOD1, SOD2, PARKIN, PINK 1, and MFN 1, astaxanthin alleviated the oxidative stress and mitochondrial dynamics failure caused by EMS. Astaxanthin restored mitochondrial oxidative phosphorylation by stimulating markers associated with the OXPHOS machinery: COX4I1, COX4I2, UQCRC2, NDUFA9, and TFAM. Our results suggest that astaxanthin has the potential to open new possibilities for potential bio-drugs to control and suppress oxidative stress, thereby improving the overall metabolic status of equine ASCs suffering from metabolic syndrome.

## 1. Introduction

Metabolic syndrome consists of various endocrine disorders encompassing insulin resistance, diabetes, obesity, and inflammation, all of which are known to substantially increase the risk of developing atherosclerotic cardiovascular disease as well as vascular and neurological complications [1]. Those diseases can develop independently, but in some cases, they may form a cluster that leads to an overall metabolic syndrome, affecting both humans and animals [2,3]. Equine metabolic syndrome (EMS) poses a particular risk to horses; it is characterized by the following factors: regional adiposity in the neck, tail head, and above the eye, insulin resistance, and laminitis, both chronic and/or acute [4]. So far, no efficient pharmacological strategy for EMS treatment has been introduced, and existing protocols are essentially based on prevention, mainly through dietary restrictions and increased physical activity [5]. Interestingly, despite regular training and a proper diet, EMS can also affect sport horses, which may indicate a more complex pathophysiological pathway [6].

One of the significant hallmarks of EMS is insulin resistance (IR) [7]. Insulin resistance is defined as a diminished biological response to insulin, a peptide hormone that regulates the anabolic response to nutrient availability by binding to receptors anchored in the plasma membrane of target cells in peripheral tissues [8,9]. IR underlines several conditions related to the metabolic syndrome, including hypertension, type 2 diabetes, and circulatory and heart diseases [10]. The metabolic role of insulin is to stimulate glucose uptake in skeletal muscle and adipocytes, promote glycogen synthesis in skeletal muscle, inhibit hepatic glucose production, and inhibit lipolysis in adipocytes [11,12].

Adipose tissue is a highly active, endocrine organ that secretes hormones and cytokines and stores energy [13]. Several studies have demonstrated that adipose tissue (AT) homeostasis is highly affected by EMS [6,14,15]. It is also considered as a key factor in overall insulin resistance and obesity development [16]. Adipose tissue is recognized as one of the main components of systemic metabolic regulation, controlling lipid mobilization and maintaining body temperature [17]. A surplus of dietary energy is stored as neutral triglycerides in adipose tissue, leading to an increase in the volume of lipid droplets and, as a result, the expansion of adipose tissue and subsequent obesity [18]. Adipose tissue, as an endocrine organ, is responsible for the synthesis and secretion of several hormones including leptin, adiponectin, visfatin, and angiotentin, which modulate insulin resistance and the inflammatory axis [6,19]. These hormones orchestrate systemic metabolism by regulating metabolically active organs such as muscles, liver, pancreas, and brain [18]. Excessive adipose tissue in horses is associated with mitochondrial dysregulation, changes in insulin signaling, increased glucocorticoid metabolism, but also changes in plasma lipid content and elevated plasma leptin levels [20]. Adipose tissue may become resistant to insulin, but due to the release of free fatty acids, pro-inflammatory cytokines, and adipokines, it also plays a key role in insulin resistance development in the whole body [21].

In EMS affected horses, AT secretes a variety of pro-inflammatory mediators, including tumor necrosis factor α (TNF-α) and interleukins [22]. Adipocytes not only synthesize and assemble triglycerides but are also able to release free fatty acids (FFA) and glycerol as triglyceride hydrolysis products [23]. High levels of FFA in the blood are strongly correlated with obesity and insulin resistance [24]. The increase in body fat leading to obesity is connected with endocrine adipocyte dysfunction [20]. Excessive caloric intake triggers an inflammatory response in adipocytes and their dysfunction through the action of cytokines such as TNFα, which reduces insulin signaling and inhibits adipogenesis [25]. Kornicka et al. demonstrated that ASC cells isolated from EMS horses have limited proliferation potential, increased senescence, apoptosis, excessive accumulation of ROS, and deterioration of the mitochondria [26]. ASC dysfunction leads to abnormal remodeling of adipose tissue, which is associated with a higher risk of metabolic disorders [27].

Recent data demonstrated that the accumulation of reactive oxygen species (ROS) paired with increased inflammation are important components that hinder the ability of ASCs to differentiate into adipocytes and limit their multipotency [28,29]. ROS are highly reactive molecules that are mainly derived from the mitochondrial electron transport chain (ETC) [30]. Cells convert molecular oxygen into superoxide anions through the monovalent reduction of molecular oxygen in ETC, respiratory burst in phagocytes resulting from the ionization of cell membrane components, and as by-products of several cellular enzymatic reactions [31]. ROS regulate some of the signaling pathways as well as cytokine secretion, proliferation, differentiation, and gene expression, and play an important role in adipogenesis [32]. However, elevated ROS levels can lead to adipocyte overgrowth and, consequently, the formation of hypertrophic adipocytes [33].

Oxidative stress is defined as the imbalance between increased levels of reactive oxygen species (ROS) and low activity of antioxidant mechanisms [34]. Increased oxidative stress damages cellular structures and may lead to acute tissue injury [35]. According to Perez-Torres et al., many natural extracts and compounds have shown valuable therapeutic potential due to their ability to efficiently scavenge various reactive oxygen species (ROS) and to prevent the activation of NF-κB and subsequent overexpression of its underlying target genes, including those involved in inflammation [36].

Carotenoids, a class of tetraterpenoids and naturally occurring pigments, have attracted great interest in the last few decades due to their potent biological activities that include antioxidant, antiproliferative, anti-inflammatory, and anti-ageing properties [37]. Astaxanthin, a red pigment usually obtained from various microorganisms and marine animals such as *Haematococcus pluvialis* algae or *Phaffia rhodozyma* yeast, emerged as a promising novel antioxidant that could be beneficial in preventing and/or reducing the risk of developing certain chronic diseases that are associated with oxidative stress-induced cellular and tissular damages [38]. Indeed, astaxanthin is able to quench and scavenge ROS and free radicals (hydrogen peroxide, superoxide anion, singlet oxygen, etc.) in both the inner and outer layers of cellular membranes, which shows its unique potential compared to most common antioxidants, which act either in the outer (vitamin C) or inner (e.g., vitamin E and β-carotene) layer of the membrane [37].

Many reports have already evidenced the beneficial pharmacological properties of astaxanthin in terms of its anti-inflammatory, immune-stimulating, anticancer, antidiabetic and antioxidant activities [39,40,41,42,43]. Astaxanthin has been demonstrated to decrease hyperglycemia-induced oxidative stress in pancreatic β-cells and to improve glucose and serum insulin levels in diabetics patients [44]. Another study reported the preventive effects of astaxanthin on a model of high glucose-induced inflammation and apoptosis in proximal tubular epithelial cells, which encourages its use for the development of new therapeutic formulations that could be applied to different pathologies and conditions [45]. At the request of the European Commission, the Panel on Nutrition, Novel Food and Food Allergens (NDA) issued statement on the safety of astaxanthin when used as a novel nutrient in food supplements at a maximum concentration of 8 mg/day [46]. In clinical studies of astaxanthin supplementation (4 mg daily) in a group of professional cyclists, an improvement of 5% in the 20 km time trial after 28 days was observed [47]. The effect of astaxanthin on skin condition was assessed in a group of 12 women receiving oral astaxanthin supplementation at a dose of 12 mg/day for 16 weeks; it was found that the treatment significantly improved skin hydration and reduced wrinkle parameters compared to the placebo group. Additionally, the levels of interleukin-1α in the stratum corneum increased significantly in the placebo group [48]. Subsequent studies have shown that oral administration of astaxanthin at a dose of 6 mg/day for 30 days alleviates the symptoms of dry eye disease in elderly patients [49]. The effect of astaxanthin supplementation at a dose of 200 mg/kg/day on cardiac diseases was assessed on a group of rats. Its administration reduced cardiomyocyte damage, inhibited inflammatory cell infiltration, preserved cardiac fiber structure, and prevented collagen deposition and stabilized levels of TGF-β1 protein in the left ventricle of high-fat rats [50]. On the other hand, dietary supplementation of astaxanthin at 0.3 mg/kg/day in healthy and obese dogs for 6 and 8 weeks, respectively, effectively activated antioxidant function and improved liver metabolic function and subsequent lipid metabolism in obese animals [51]. Collectively, these data suggest that astaxanthin may represent a novel drug candidate for the proper management of metabolic disturbances not only in humans but also in veterinary medicine. In the present study, we hypothesize that astaxanthin could be a beneficial therapeutic lead for restoring the metabolic balance in EMS adipose-derived stromal stem cells by targeting aberrant oxidative stress and the underlying mitochondrial dysfunction. For this purpose, we assessed the effects of astaxanthin on cell viability, apoptosis, reactive oxygen levels, mitochondrial dynamics, and metabolism, as well as its antioxidant capacity. The obtained results suggest that astaxanthin improves the metabolic status of equine ASC affected by metabolic syndrome. This finding opens up new possibilities for creating potential biopharmaceuticals.

## 2. Materials and Methods

### 2.1. Yeast Biomass and Astaxanthin Extraction

*Phaffia rhodozyma* NCYC 874 was obtained from the National Collection of Yeast Culture (UK) and cultured in Yeast Extract-Peptone-Dextrose medium (Sigma Aldrich; Poznań, Poland) for 72 h at 21 °C in a shaker at 160 rpm (LS 500 POL-EKO Aparatura; Wodzisław Śląski, Poland). Cells were afterwards collected by centrifugation at 3200× *g* for 5 min, 4 °C.

The biomass (5 g) was extracted with 100 mL of acetone (Sigma Aldrich; Poznań, Poland) in a shaker at 160 rpm at 30 °C for 2 h. After pelleting the yeast cells by centrifugation at 3200× *g* for 5 min, 4 °C the solvent was evaporated in a ventilated incubator completely from the sample at 35 °C in the dark and stored at 4 °C in the dark until use.

### 2.2. Equine ASCs Cell Culture

Equine ASCs were obtained from the cell collection of the Department of Experimental Biology, University of Environmental and Life Sciences, Wroclaw, Poland. Stem cells population expressed both CD90 and CD105 markers and were negative for CD45 and CD34 which excluded their hematopoietic origin [52].

ASCs cultures were maintained in 75 cm^2^ flasks and cultured in Dulbecco’s modified Eagle’s medium (DMEM) containing 1000 mg/L glucose, supplemented with 5% of fetal bovine serum (FBS), and 1% of a penicillin and streptomycin (PS) solution, and incubated at 37 °C in a humidified 5% CO_2_ incubator. Cultured cells were harvested every three days (80–90% of confluence) and detached from the flasks with a trypsin-EDTA solution (TrypLE Express, Life Technologies; Carlsbad, CA, USA). Cells were multipotent and able to differentiate into adipocytes, chondrocytes and osteoblast in vitro as in previous study [53].

### 2.3. Determination of Cell Viability and Proliferative Activity by TOX8 Assay

The effect of astaxanthin on the viability and proliferation of equine ASC cells was assessed using a Resazurin-based assay kit (TOX8). Briefly, cells were seeded onto 96-well plate at a density of 8 × 10^3^ cells/well in a final volume of 100 µL of DMEM complete culture medium. Next, cells were treated with 1, 5, 10, 20, 50 and 100 μg/mL of astaxanthin for 24, 48 and 72 h. After each incubation, the remaining medium was removed, and 100 µL of a 10% resazurin solution were added to each and incubated for 2 h at 37 °C in a humidified 5% CO_2_ incubator. Absorbance was measured spectrophotometrically (SPECTROstar Nano, BMG LABTECH, Ortenberg, Germany) at a wavelength of 690 nm as a reference wavelength and 600 nm for resazurin. Two optimal concentrations of astaxanthin, i.e., 10 and 20 μg/mL, were selected for further experiments.

### 2.4. Bromodeoxyuridine (BrdU) Assay

The effects of astaxanthin on the DNA synthesis and proliferation were assessed using the 5-bromo-2-deoxyuridine (BrdU) Cell Proliferation ELISA Kit (Abcam, Cambridge, UK) according to the manufacturer’s instructions. Briefly, after 24 h treatment of EqASC cells with astaxanthin at concentrations of 10 and 20 μg/mL, BrdU reagent was added to all culture media and incubated at 37 °C during 72 h. After fixation of the cell, the detection of the incorporated BrdU was performed using anti-BrdU primary monoclonal antibody and secondary goat anti-mouse IgG conjugated with horseradish peroxidase (HRP). HRP substrate degradation was measured using a plate reader spectrophotometer (Spectrostar Nano; BMG Labtech, Ortenberg, Germany) at a 450 nm wavelength.

### 2.5. Colony-Forming Unit-Fibroblast (CFU-fs) Assay

The effect of astaxanthin on the ability of cells to form colonies was assessed by seeding 100 EqASC cells (HE and EMS) per well in six-well plates. Astaxanthin was subsequently added to the culture medium at different concentrations of 10 and 20 μL/mL and incubated for 7 days at 37 °C and 5% CO_2_. The cells colonies were fixed in 4% cold paraformaldehyde and stained with pararosaniline solution. Colonies containing more than 50 cells were counted using an inverted microscope (AxioObserverA1; Zeiss, Oberkochen, Germany). The calculation of the efficiency of colony forming (CFU) was based on the following formula:CFU-fs (%) = (Number of colonies/Initial cell number) × 100.(1)

### 2.6. Flow Cytometric Analysis of Cell Viability and Apoptosis

The percentage of viable and apoptotic cells were determined using the MUSE™ Annexin V & Dead Cell Kit (Merck Millipore, Darmstadt, Germany) according to the manufacturer’s instructions. Briefly, after treatment with astaxanthin, all treated and untreated EqASC cells were collected by trypsinization and suspended in Hanks’ Balanced Salt Solution (HBSS) containing 1% FBS. Then, cells were stained with the Annexin V & Dead Cell reagent for 20 min at room temperature and analyzed using the Muse Cell Analyzer (Merck Millipore, Darmstadt, Germany). The cell survival and apoptosis ratio were calculated by the identification of four distinct populations: (i) non-apoptotic cells, not undergoing detectable apoptosis, i.e., Annexin V (−) and 7-AAD (−); (ii) early apoptotic cells, Annexin V (+) and 7-AAD (−); (iii) late apoptotic cells, Annexin V (+) and 7-AAD (+); and (iv) cells that had died through non-apoptotic pathway, i.e., Annexin V (−) and 7-AAD (+).

### 2.7. Intracellular Reactive Oxygen Species Determination

The accumulation of intercellular reactive oxygen species (ROS) was evaluated using the Muse^®^ Oxidative Stress Kit (Merck Millipore, Darmstadt, Germany) according to the manufacturer’s instructions. Briefly, untreated and treated EqASC cells were collected and washed with HBSS. The cells were resuspended in the Muse Oxidative Stress working solution and incubated for 30 min at 37 °C in the dark. The measurement of ROS+ versus ROS− populations was performed by a Muse Cell Analyzer (Merck Millipore, Darmstadt, Germany).

### 2.8. Endogenous Antioxidant Activities Assays

The activity of endogenous antioxidant enzymes was evaluated after the collection of lysed cells by sample assay buffer and centrifugation at 1200× *g* for 10 min. The antioxidant capacity was measured using the Cayman’s Antioxidant Assay Kit (Cayman Chemical Company, Ann Arbor, MI, USA) following the manufacturer’s instructions. Briefly, samples were mixed with 10 μL of metmyoglobin and 150 μL of chromogen in a 96-well plate and compared to 10 μL of Trolox standard at different concentrations. Then 40 μL of hydrogen peroxide was added to each plate and incubated for 5 min at room temperature. Data analysis was performed using a spectrophotometer plate reader (Spectrostar Nano; BMG Labtech, Ortenberg, Germany) at 750 nm. The activity of superoxide dismutase (SOD) was measured using the Cayman Superoxidase Dismutase Assay Kit (Cayman Chemical Company, Ann Arbor, MI, USA) according to the instruction in the user’s guide. Briefly, 200 μL of radical detector was mixed with 10 μL of each sample on standard concentration in a 96-well plate. Then 20 μL of xanthine oxidase was added to each well and incubated for 30 min in room temperature and read immediately at 460 nm using spectrophotometer plate reader (Spectrostar Nano; BMG Labtech, Ortenberg, Germany).

Results were expressed as Trolox Equivalent Antioxidant Capacity (TEAC) in mM and activity percent for SOD.

### 2.9. Mitochondrial Membrane Potential Assay (MMP)

Measurement of changes in mitochondrial membrane potential (ΔΨm) was performed with the MUSE™ MitoPotential Assay kit (Merck Millipore, Darmstadt, Germany). After treatment of the cells with astaxanthin, the culture medium was removed and EqASC cells were washed twice with HBSS, then incubated with the MitoPotential fluorescent dye for 30 min at 37 °C. The percentage of depolarized cells (depolarized alive + depolarized dead) was determined by Muse™ Cell Analyzer (Merck Millipore, Darmstadt, Germany).

### 2.10. Mitochondrial Network Fluorescent Staining

To visualize the morphology of the mitochondria, cells were stained with MitoRed (Sigma Aldrich, Poznan, Poland) fluorescent dye (1:1000 in culture medium), following each related treatment, and incubated for 30 min at 37 °C, prior to fixation in 4% paraformaldehyde at room temperature for 45 min. ProLong™ Diamond Antifade Mountant with DAPI (Invitrogen™, Warsaw, Poland) was used for nuclei staining. The stained cells were observed using a confocal microscope (Observer Z1 Confocal Spinning Disc V.2 Zeiss with live imaging chamber) and captured with a Canon PowerShot camera. Obtained photomicrographs were merged and analyzed using ImageJ software (Bethesda, Rockville, MD, USA). Differences in mitochondria morphology were further evaluated using the Imaris software (Imaris^®^, Bitplane AG, Oxford Instruments, Zürich, Switzerland).

### 2.11. Mitochondria Isolation for Transcriptomic Analysis

To investigate if the treatment of astaxanthin influenced mitochondrial metabolic changes, total mitochondria were isolated from EqASC cells using the commercial Thermo Scientific™ Mitochondria Isolation Kit for Cultured Cells (Thermo Fisher Scientific, Warsaw, Poland), according to the manufacturer’s instructions. Briefly, all treated and untreated cells were collected from culture flasks and washed three times with cold HBSS, then centrifuged at 300× *g* for 4 min, at 4 °C. remaining cells pellets were sequentially lysed using the provided lysis reagents containing a protease inhibitors cocktail (1:1000) on ice. The remaining cellular derbies and cytosolic fractions were discarded by centrifuging cell lysates at 700× *g* for 10 min at 4 °C. The obtained total mitochondria-rich supernatants were subsequently centrifuged at 12,000× *g* for 15 min at 4 °C. The final mitochondria pellets were then resuspended in TRIzol reagent for mtRNA isolation.

### 2.12. RNA Extraction and Real-Time Reverse Transcription PCR (qRT-PCR)

Total RNA was extracted from EqASC cells using TRIzol reagent according to the manufacturer’s instructions. RNA purity and concentration were measured using a nanospectrophotometer (WPA, Biowave II, Cambridge, UK). cDNA was prepared from total isolated RNA using a PrimeScript™ RT Reagent Kit with gDNA Eraser (TaKaRa, Gdańsk, Poland) by the mean of a T100 Thermal Cycler (Bio-Rad, Hercules, CA, USA) according to the manufacturer’s instructions.

Mt-cDNA samples were preamplified prior to quantitative RT-PCR analysis using a pool of primers based on targeted mitochondrial-related genes as in the cycling conditions: 95 °C for 2 min, followed by 18 cycles at 95 °C for 3 s, annealing for 3 min and 72 °C for 3 s.

The gene expression levels were evaluated by real-time reverse transcription polymerase chain reaction (RT-qPCR) using SensiFAST SYBR Green Kit (Bioline, London, UK) in a CFX Connect™ Real-Time PCR Detection System (Bio-Rad). Briefly, 10 μL total volume of each reaction consisted of 5 μL of SensiFAST SYBR Master mix, 2.5 μL of targeted primer and 2.5 μL of tested cDNA. Thermal cycle conditions were as follows: 95 °C for 2 min, then 40 cycles at 95 °C for 15 s, annealing for 15 s in temperature specified for tested primers, and elongation at 72 °C for 15 s. The results were reported regarding the expression of the housekeeping glyceraldehyde 3-phosphate dehydrogenase (GAPDH). The relative gene expression was calculated for all the tested groups, i.e., healthy, EMS, and EMS treated with astaxanthin using the 2^−ΔΔCQ^ method. The sequences for all used primers are listed in Table 1.


### 2.13. Western Blot Analysis

The EqASC cells were collected from each culture flask and homogenized in a mixture of phosphatase, protease inhibitor, and lysis buffer (Tris at 50 mmol/L pH 7.4, NaCl at 150 mmol/L, 0.1% SDS, 0.5% sodium deoxycholate, protease cocktail, 1 mmol/L PMSF, 10 mmol/L sodium ascorbate, 1% Triton X-100, 10 mmol/L of sodium azide, and Trolox at 5 mmol/L) on ice, in order to perform protein profiling. Proteins were collected by centrifugation of cell lysates for 20 min at 4 °C and 6000× *g* to remove insoluble materials, and subsequently transferred to new 1.5 mL Eppendorf tubes. Protein concentration was determined using the Pierce™ Bicinchoninic Acid (BCA) Protein Assay Kit. SDS-polyacrylamide gel electrophoresis was performed for 90 min in Tris/glycine/SDS 100 V buffer using a Mini-PROTEAN Tetra vertical electrophoresis cell (Bio-Rad, Hercules, CA, USA), on samples diluted in 4 × Laemmli Loading Buffer (Bio-Rad, Hercules, CA, USA) and denatured at 95 °C for 5 min. Then protein transfer was performed using polyvinylidene difluoride (PVDF) membranes (Bio-Rad, Hercules, CA, USA) with a Mini Trans-Blot^®^Cell transfer apparatus (Bio-Rad, Hercules, CA, USA) in Tris/glycine/methanol buffer with 100 V, 250 mA at 4 °C for 45 min. Protein membranes were blocked in 5% skim milk solution in TBST for 1 h at room temperature. Protein detection was performed by incubation overnight at 4 °C in primary antibodies (Table 2), and secondary antibodies conjugated to HRP, dilution 1: 2500 in TBST, for 1 h at room temperature. Chemiluminescent signals were acquired using the ChemiDoc MP imaging system (Bio-Rad, Hercules, CA, USA) and quantified by Image Lab software (Bio-Rad, Hercules, CA, USA).

### 2.14. Statistical Analysis

Statistical analyses were performed using the GraphPad Prism 8.0 (San Diego, CA, USA). Statistical significance was determined using one-way analysis of variance (ANOVA) with Dunett’s post hoc multiple comparison test. Asterisk (*) signs indicated statistical significance of EqASCs EMS control versus EqASCs healthy cells and EqASCs EMS control versus EqASCs EMS-Astaxanthin treated groups respectively. The *p* values lower than 0.05 (*p* < 0.05) were summarized with one asterisk (*), *p* < 0.01 with two asterisks (**), and *p* < 0.001 with three asterisks (***).

## 3. Results

### 3.1. Astaxanthin Improves Viability and Proliferation in EMS ASCs Affected Cells

The impact of astaxanthin on cell proliferation was evaluated in terms of metabolic activity, DNA synthesis, and colony forming assays. As illustrated in Figure 1a, EMS affected cells were characterized by considerably lowered metabolic activity, as evidenced by the reduced ability to efficiently metabolize the Resazurin blue dye, in comparison to healthy cells after 24, 48, and 72 h of incubation. The obtained results also revealed that astaxanthin had no cytotoxic effect on EqASC cells at all tested concentrations (1, 5, 10, 20, 50, and 100 µg/mL). What is more, after 48 and 72 h of incubation, astaxanthin significantly increased the proliferation capacity of EMS EqASC cells in comparison to untreated cells (*p* < 0.001) (Figure 1a). Similar trends were observed after BrdU incorporation analysis, which revealed that EMS-affected ASCs had decreased newly synthetized DNA and thus reduced proliferation potential, by contrast to healthy untreated cells (Figure 1b). Treatment of EMS affected cells with astaxanthin at 10 and 20 g/mL resulted in a significant improvement in cell proliferation and division, as demonstrated by the increased amount of newly synthesized DNA in opposition to untreated EMS cells control group (*p* < 0.01). Analogously, the clonogenic fibroblast precursor (CFU-F) assay revealed a significant increase in the number of cells colonies formed by EMS EqASC cells after astaxanthin treatment at the two optimal tested concentrations (Figure 1c), whereas untreated cells had a lower ability to form colonies when compared to healthy control cells (*p* < 0.001). Astaxanthin promoted cell proliferation and division, and the proportion of CFU-Fs produced in the astaxanthin-treated groups was equivalent to that observed in normal ASC cells.

### 3.2. Astaxanthin Reduced Cell Apoptosis in Equine ASC Cells Suffering from EMS

The proportion of viable cells versus cells undergoing apoptosis following astaxanthin treatment was determined using the Muse^®^ Annexin V & Dead Cell test, as well as by measuring the gene expression of apoptosis master regulators. In contrast to healthy, non-affected cells, EMS cells were prone to increased apoptosis, as demonstrated by a drop in the percentage of live cells, and a resulting increase in the proportion of apoptotic and dead cells (Figure 2a). EMS ASC cells treated with two different concentrations of astaxanthin (10 and 20 µg/mL), displayed a significantly lower number of dead cells when compared to EMS untreated group (Figure 2b), suggesting that astaxanthin has a beneficial effect on cell viability. Furthermore, treated cells had low to moderate percentages of total apoptotic and dead cells, indicating that the carotenoid exerts an anti-apoptotic impact. Similarly, the relative expression of pro- and anti-apoptotic markers was assessed at mRNA level using the RT-qPCR approach. Obtained data clearly demonstrated that EMS cells were characterized by a considerable elevation of key pro-apoptotic factors expression, including *p21*, *p53*, *Bax*, *Casp-3*, *Casp-8*, and *Casp-9* in comparison to healthy control cells (Figure 2c), while apoptosis inhibitor *Bcl-2* appeared to be markedly downregulated in EMS cells (*p* < 0.05).

Treatment of EMS ASC cells with astaxanthin enabled the significant reduction of the excessive apoptosis by suppressing *p21*, *p53*, *Bax*, *Casp-3*, *Casp-8*, and *Casp-9* overexpression while simultaneously improving *Bcl-2* pro-survival transcript expression (Figure 2c), implying that astaxanthin has a strong anti-apoptotic and pro-survival effect on EMS ASC cells.

### 3.3. Astaxanthin Decreases Oxidative Stress in Equine EMS ASC Cells

Oxidative stress represents one of the most prominent hallmarks of EMS in ASC cells, leading to global metabolic failure, and ultimately, to cellular death. In this study, the Muse Oxidative Stress Assay was used to assess intracellular ROS accumulation in untreated and astaxanthin-treated cells (Figure 3a). A significant rise in intracellular ROS in EMS ASC cells was found (Figure 3b), confirming the initiation of oxidative stress during EMS. The treatment with astaxanthin over a period of 24 h effectively abolished the relevant EMS-associated oxidative stress, as demonstrated by the decrease in the number of ROS positive cells in comparison to untreated EMS cells (*p* < 0.001). The antioxidant effect of astaxanthin was further investigated by measuring SOD enzyme activity and total cellular antioxidant capacity (Figure 3c). In ASC cells affected by EMS, the collected data revealed a substantial decrease in SOD activity and total endogenous antioxidant capacity (*p* < 0.001). Treatment with astaxanthin resulted in a substantial restoration and stimulation of the cellular antioxidant capacity, together with improved enzymatic efficacy of SOD under EMS condition. This effect was further confirmed by RT-qPCR results, which showed that EMS affected cells had dysregulated *Sod1* and *Sod2* gene expression (Figure 3d), whereas EMS ASC cells treated with astaxanthin displayed an upregulation and recovery of the same transcripts, indicating that astaxanthin improved the antioxidant status of affected cells, resulting in oxidative stress mitigation.

### 3.4. Astaxanthin Enhances Mitochondrial Dynamics in EMS Affected ASC cells

Mitochondrial metabolism disruption is a key player in oxidative stress, cellular ROS overproduction, and resulting pro-apoptotic pathways activation, which are characteristic of EMS. The significant increase in total living and dead cells exhibiting depolarized mitochondrial membrane potential in EMS untreated ASC cells (Figure 4a,b), in contrast to metabolically normal cells (*p* < 0.05), revealed that the EMS condition caused mitochondrial activity collapse. In addition, there was evidence of a significant breakdown in mitochondrial dynamics. An analysis of the morphology of the mitochondria highlighted an evident mitochondrial tubular network disruption, as EMS untreated cells exhibited poor and fragmented mitochondrial networks, characterized by reduced branched tubular and globular structures and critical loss in mitochondrial loops (Figure 4c–e), suggesting the depletion of fusion capacity in favor of sustained fission. In fact, EMS untreated cells showed significant downregulation of mitochondrial fusion related factors, such as *MFN-1* and *OPA-1* (Figure 4f), while the expression of fission associated markers, including *Parkin* and *Fis-1* (*p* < 0.01), appeared to be significantly higher than that of healthy control cells. Western blot analysis further showed increased MFF and Pink1 protein expression, two major regulators of mitochondrial fission in EMS untreated ASC group (Figure 4g) compared to control cells (*p* < 0.01), confirming the aberrant and persistent mitochondrial fragmentation and division under EMS condition. Twenty-four hours of astaxanthin conditioning resulted in a substantial improvement of overall mitochondrial function (Figure 4). In fact, EMS ASC cells treated with astaxanthin were characterized by enhanced membrane potential, as evidenced by the decrease in the percentage of cells with depolarized mitochondria (Figure 4b).

Compared to EMS-untreated EqASC cells, astaxanthin-treated cells further showed mild mitochondrial dynamics perturbations, as demonstrated by the positive regulation of fusion/fission balance and the consequent suppression of excessive *Parkin* and *Fis-1* transcripts expression, the downregulation of MFF and Pink1 proteins levels, as well as the restoration of *MFN-1* and *OPA-1* transcription (Figure 4f). As a result, a restoration of the mitochondrial network architecture with promoted globular structures, branched tubular shape, and triple-stranded loop distribution in carotenoid-treated cells was observed (Figure 4c–e), all of which suggest that astaxanthin may lower oxidative stress and apoptosis by rebalancing mitochondrial dynamics.

### 3.5. Astaxanthin Supports the Transcription of Mitochondrial Metabolism Related Effectors

Mitochondria are considered central metabolic hubs that maintain overall energetic homeostasis through respiration and electron transfer reactions. To further support the evidence that astaxanthin may exert its antioxidant effect through the amelioration of mitochondrial activity and metabolism, an analysis of the expression of genes linked to mitoribosomes and mitochondrial oxidative phosphorylation machinery (OXPHOS) in all experimental groups of cells was carried out using RT-qPCR. The obtained data (Figure 5a) showed that EMS cells displayed diminished *Uqcrc2*, *Ndufa9*, *Cox4l1*, and *Cox4l2* gene expression as compared to non-stressed cells. Furthermore, master regulators of mitochondrial transcription machinery as well as mitochondrial ribosomal biogenesis and translation, namely *Pusl1*, *Mrpl24*, and *Tfam*, were profoundly downregulated under EMS condition, which indicated a severe failure in mitochondrial biogenesis and oxidative phosphorylation (Figure 5b). Moreover, the *Wnt3* transcript, which is involved in the upregulation of the OXPHOS complex, was found to be significantly downregulated in EMS affected cells, suggesting a disruption of the mitochondrial metabolic regulation pathways. The addition of astaxanthin to EMS ASCs cell cultures for a period of 24 h significantly improved overall mitochondrial metabolism, as evidenced by a visible recovery of the expression of OXPHOS associated complexes units, including *Uqcrc2*, *Ndufa9*, *Cox4l1*, and *Cox4l2*, together with the restoration of factors involved in mitochondrial translation processes, including *Pusl1*, *Mrpl24*, and *Tfam*, as well as a member of the *Wnt/β-catenin Wnt3* signaling pathways, which exert various regulatory actions in cellular energy metabolism (Figure 5). These results indicate the likely specific antioxidant effect of astaxanthin, i.e., it may target mitochondrial metabolism and its associated oxidative phosphorylation machinery.

## 4. Discussion

In the present study, we investigated the effects of astaxanthin on preventing apoptosis, mitochondrial dysfunction and oxidative stress of adipose-derived stromal stem cells affected by EMS. Astaxanthin is a promising bioactive compound for the prevention of several human diseases as well as for the maintenance of good health [41]. Astaxanthin exhibits a wide range of biological functions, most of which are associated with its antioxidant and anti-inflammatory effects [54]. The growing popularity of astaxanthin is supported by its high ability to absorb oxygen radicals, i.e., 100 to 500 times higher than that of *α*-tocopherol, and 10 times more active inhibition of free radicals than related antioxidants (*α*-tocopherol, *α*-carotene, *β*-carotene, lutein and lycopene) [55]. Astaxanthin reduces cell apoptosis, ameliorates oxidative stress and mitochondrial dysfunction and modulates mitochondrial dynamics [56,57]. This study provides strong evidence for the role of astaxanthin in equine metabolic syndrome prevention and treatment.

Our research has shown that EMS affects ASC viability and correlates with oxidative stress. EMS contributes to insulin resistance and the secretion of various adipokines, which worsens the metabolic state of the body [58]. Our studies on the effect of astaxanthin on ASC (healthy and EMS controls) showed an improvement in cell viability without cytotoxic effect, mainly at concentrations of 10 and 20 μg/mL. Moreover, the results showed that treating EMS ASCs with two different concentrations of astaxanthin significantly increased their proliferation rates compared to untreated EMS cells, which had reduced proliferation and colony formation capacity and a high apoptotic tendency. Research by Weiss et al. has shown an over-activation of apoptotic pathways in ASC cells affected by EMS, with a consequent reduction in proliferation rate and an increase in cell death rate [59]. These results correlate with our studies, which showed an increased number of apoptotic cells accompanied by critical upregulation of pro-apoptotic players including *p53*, *p21*, *Bax*, *Casp-3*, *Casp-9* and *Casp-8* transcripts, in parallel with a loss of expression of the *Bcl-2* cell survival gene. According to Cui et al., astaxanthin is a powerful anti-apoptotic agent which prevents ochratoxin A-induced heart damage and cardiomyocyte apoptosis in mice, mainly by regulating the expression of *Keap1*, *Nrf2*, *Bax*, *Bcl-2*, *Casp-3* and *Casp-9* both at the mRNA and protein level [60]. Guo et al. reported that astaxanthin significantly stimulated phosphorylation in an experimental model of acute kidney injury induced by burns in rat, which then allowed the inhibition of activation of further pro-apoptotic factors, including cytochrome c and *caspase-3/9* axis [61]. The above reports support our results, where the use of astaxanthin in EMS ASC cultures for 24 h resulted in a clear stimulation of the metabolic activity and proliferation rate of cells, along with the promotion of their ability to create colonies, while preventing increased apoptosis by suppressing *p53*, *p21*, *Bax*, *Casp-3*, *Casp-9* and *Casp-8* expression and simultaneously restoring *Bcl-2* survival gene expression.

Oxidative stress disturbs cell homeostasis because of an excess of free radicals in relation to the number of antioxidant molecules [62]. An excess of generated ROS can have a negative impact on cell development, lipid metabolism, nucleic function, cellular communication and control, genetic mutations and biological activity, immune activation and inflammation [63]. Oxidative stress is involved in the development of insulin resistance and coexisting inflammation [64]. Additionally, it has been shown that oxidative stress in obese patients with insulin resistance and hyperglycemia increases the risk of aggressive cardiovascular diseases and pro-inflammatory changes and reduces the bioavailability of nitric oxide [65]. In this study, we confirmed that EMS ASC was characterized by increased intracellular accumulation of ROS due to the onset of severe oxidative stress and decreased expression of the antioxidant enzymes SOD1 and SOD2. Simultaneously, EMS ASCs showed a significant decrease in total endogenous antioxidant capacity and an associated decreased SOD enzymatic activity. EMS is strongly linked to insulin resistance; it affects an increasing number of horses and has already been reported to exhibit strong and excessive oxidative stress, which strongly impairs metabolic functions and accelerates cellular senescence and ageing, leading to multipotency limitation [66]. ROS play a significant role in diseases related to metabolic dysregulation and inflammation; therefore, EMS ASC cells affected by prolonged oxidative stress are characterized by an overproduction of harmful ROS, accompanied by the breakdown of cellular antioxidant defense mechanisms, including a decrease in the activity of antioxidant enzymes and their expression at the gene level [6,67]. According to our results, astaxanthin modulates oxidative stress within ASC cells; this was evidenced by the observed drop in the number of ROS positive cells and the restoration of the expression of the two antioxidant enzymes (SOD1 and SOD2). Astaxanthin has a strong antioxidant effect due to the presence of two oxidized groups on each ring, so it can regulate key signaling pathways by regulating or activating various molecules and pathways, such as *PI3K/AKT* and *JAK/STAT-3* (signal transducers and transcription activators), *Nrf2* (NF-E2 related factor 2), *NF-κB* (kappa nuclear factor-activated B-cell light chain enhancer), MAPK (mitogen-activated protein kinases) and PPARγ (peroxisome proliferator-activated gamma receptor) [68]. Astaxanthin showed remarkable protective effects on cellular membranes and associated lipid peroxidation due to its polar/non-polar chemical structure [69]. In addition, it has been proven that astaxanthin has a strong effect on the reduction of lipid damage in liposomes treated with H_2_O_2_, tert-butyl hydroperoxide (t-ButOOH) or ascorbate and Fe^2+^: EDTA [70]. Research by Cui et al. and Xue et al. showed that astaxanthin reduces the levels of ROS which are directly related to LDL oxidation and overall lipid peroxidation. The observed effects were related to the stimulation of expression and mobilization of the nuclear factor 2-related transcription factor 2 (*Nrf2*), followed by upregulation of its target antioxidant genes, including phase II biotransformation enzymes [60,71].

Our research has shown changes in mitochondrial membrane potential, which is a key marker of mitochondrial dysfunction and is considered an early indicator of cellular stress and apoptosis. In addition, a morphological analysis showed a loss of mitochondrial network integrity, as evidenced by excessive cleavage and decreased fusion, which was further confirmed by the observed dysregulation of the axis of fusion/cleavage markers, including *Parkin*, *Fis-1*, *MFN-1* and *OPA-1*. The performed analysis of the integrity of mitochondrial biogenesis showed a breakdown in mRNA expression of *Mrpl24*, *Mterf4*, *Tfam* and *Pusl1*, major mtDNA regulators and key players in the biogenesis of mitochondrial ribosomes and translation machines. Our report correlates with the research of Marycz et al., where ASC EMS was shown to suppress gene expression related to selective mitophagy and mitochondrial dynamics and biogenesis, including *Pink*, *Parkin*, *PGC1-α* and *PDK4* [72]. Initially, astaxanthin has an antioxidant effect, eliminating reactive oxygen and nitrogen species and other free radicals [73]. However, there are more and more reports on the stimulation of the action and function of mitochondria in cells treated with astaxanthin [71,74,75,76,77]. According to our research, cells treated with astaxanthin showed increased mitochondrial dynamics and biogenesis, as demonstrated by restoring mitochondrial transmembrane potential, restoring proper mitochondrial morphology, and regulating fission events and fusion dynamics. Research of Nishid et al. also confirmed the beneficial effect of astaxanthin on the maintenance and enhancement of mitochondrial activity through a direct impact on the *AMPK/Sirtuin/PGC-1α* pathway and other pathways [56]. Nawaz et al. confirmed the ability of astaxanthin to stimulate mitochondrial function, where astaxanthin was shown to reduce insulin resistance in diet or myotubule-induced obesity in vitro by modulating insulin signaling in an antioxidant and antioxidant-independent manner and by activating mitochondrial energy metabolism via the activation of the AMP-activated protein kinase pathway (AMP) coactivator *γ* coactivator-1*α* (PGC-1*α*) peroxisome proliferator activated receptor in skeletal muscle [56]. Yu et al. proved that astaxanthin maintained mitochondrial integrity by increasing *PGC-1α* expression and maintained normal tubular structure and heat-induced oxidative stress in C2C12 myoblasts [78]. Additionally, Jiang et al. showed that astaxanthin induced the expression of genes encoded in mitochondria, increased the number of viable copies of mitochondria and stimulated the activity of mitochondrial respiratory chain complex enzymes in BPA-damaged rat kidneys [79].

Cellular energy mainly results from the production of ATP via oxidative phosphorylation (OXPHOS) [80]. The tightly folded inner membranes of mitochondria called cristae contain OXPHOS (I-IV) complexes, including ATP synthase (V complex), while complexes I-IV are multi-subunit enzymes that work together to form an electrochemical proton gradient in the mitochondrial inner membrane that is used for the production of ATP by oxidative phosphorylation [81]. The OXPHOS system may trigger the development of several cellular disorders, including inflammation, oxidative stress and apoptosis [82]. Here, we showed for the first time that equine ASC cells affected by EMS presented a serious failure in the OXPHOS machinery, as evidenced by the substantial downregulation of the *Ndufa9, Uqcrc2, Cox4i1* and *Cox4i2* transcripts, suggesting that one of the molecular mechanisms leading to oxidative stress and mitochondrial failure during EMS lies in the affliction of the OXPHOS system and downstream disruption of mitochondrial metabolism and cellular energy homeostasis. Moreover, *Wnt3* gene expression, which has been implicated in the regulation and activation of the OXPHOS action, appeared to be significantly compromised, indicating a concomitant loss of regulatory pathways. Our outcomes established that astaxanthin also targets mitochondrial OXPHOS complexes by upregulating the expression levels of each related complex mRNA together with *Wnt3* regulator, supporting the hypothesis that astaxanthin’s antioxidant properties may involve a restoration of proper oxidative phosphorylation.

Currently, to the best of our knowledge, there are no published clinical studies on astaxanthin supplementation in horses suffering from EMS. Nevertheless, studies by Sato et al. showed that supplementation of astaxanthin and L-carnitine during training of Thoroughbred horses reduced the incidence of exercise-induced muscle damage due to their mutual antioxidant properties [83]. Research on the use of natural extracts against EMS was conducted by Nawrocka et al. Those authors showed that a 3-month supplementation of *Spirulina plantesis*, rich in phycocyanin and β-carotene, reduced body mass, as well as improving serum insulin levels and sensitivity in the group of horses with EMS [84]. In a study on a rat model by Hussein et al., supplementation with astaxanthin at a dose of 50 mg/kg/day for 22 weeks lowered blood pressure and fasting blood glucose levels, as well as insulin resistance index, which subsequently restored insulin sensitivity and overall metabolic homeostasis [85]. Zhuge et al. also investigated the anti-diabetic effects of astaxanthin in male diabetic rats who were fed a diet containing 15 mg/kg/day of astaxanthin for 3 weeks. Supplementation with astaxanthin significantly lowered the level of blood glucose and total cholesterol (TC) and increased the level of high-density lipoprotein (HDL-C) cholesterol [86]. In view of the above reports, it would be reasonable to conduct clinical trials on astaxanthin supplementation in horses with EMS in order to confirm whether Astaxanthin exhibits similar in vivo antioxidant and antidiabetic effects.

## 5. Conclusions

According to the presented results, astaxanthin showed promising abilities for the treatment of metabolically afflicted ASC cells, and prevented the occurrence of apoptosis, reduced oxidative stress, and reversed mitochondrial dysfunctions, which are key players in the development of EMS condition and are known to seriously impair the regenerative properties of ASC cells. Furthermore, through the modulation of mitochondrial dynamics and OXPHOS, astaxanthin is providing new insights in the treatment of insulin resistance and obesity. Additional indepth study would provide more precisions regarding the detailed mechanisms by which Astaxanthin improves mitochondrial biogenesis and thus ameliorates the fate of malfunctioning EMS ASCs.

## Figures and Tables

**Figure 1 biomolecules-12-01039-f001:**
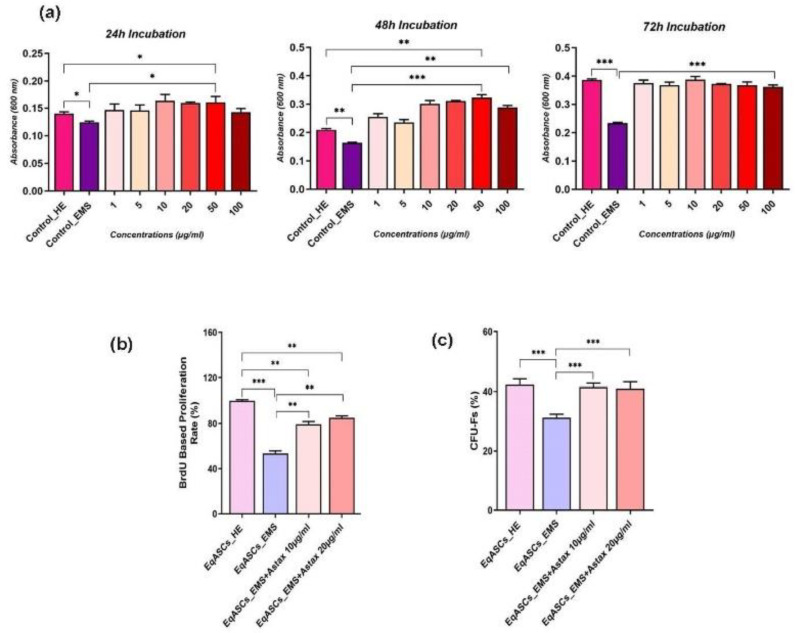
Impact of astaxanthin on cell proliferation. (**a**) The average absorbance at 600 nm after 24 h, 48 h, and 72 h of metabolized resazurin dye by healthy, treated, and untreated cells are shown by histograms. (**b**) Percentage of incorporated BrdU in newly synthetized DNA. (**c**) Cell proliferation assay using the clonogenic fibroblast precursor (CFU-F) assay. A comparison of EMS, treatment groups and untreated healthy cells is shown by an asterisk (*). * *p* < 0.05, ** *p* < 0.01, *** *p* < 0.001.

**Figure 2 biomolecules-12-01039-f002:**
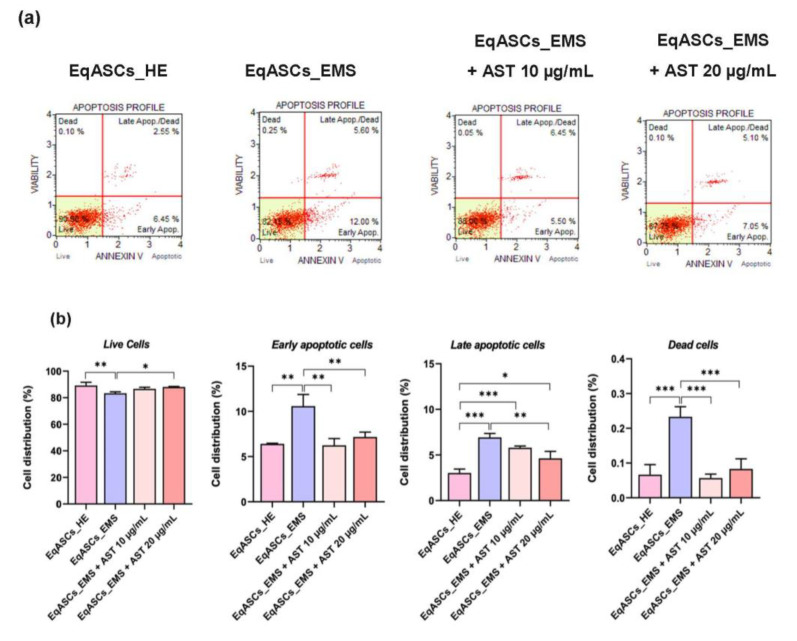
Antiapoptotic effect of Astaxanthin on EMS ASC cells. (**a**) The Muse^®^ Annexin V & Dead Cell assay was used to assess live cells, early and late apoptotic cells, and dead cells. (**b**) According to the Muse^®^ Annexin V & Dead Cell assay, histograms reflect the ratio of live, early apoptotic, late apoptotic, and dead cells. (**c**) Bar charts illustrating the relative expression of major apoptotic markers. A comparison of EMS, treatment groups and untreated healthy cells is shown by an asterisk (*). * *p* < 0.05, ** *p* < 0.01, *** *p* < 0.001.

**Figure 3 biomolecules-12-01039-f003:**
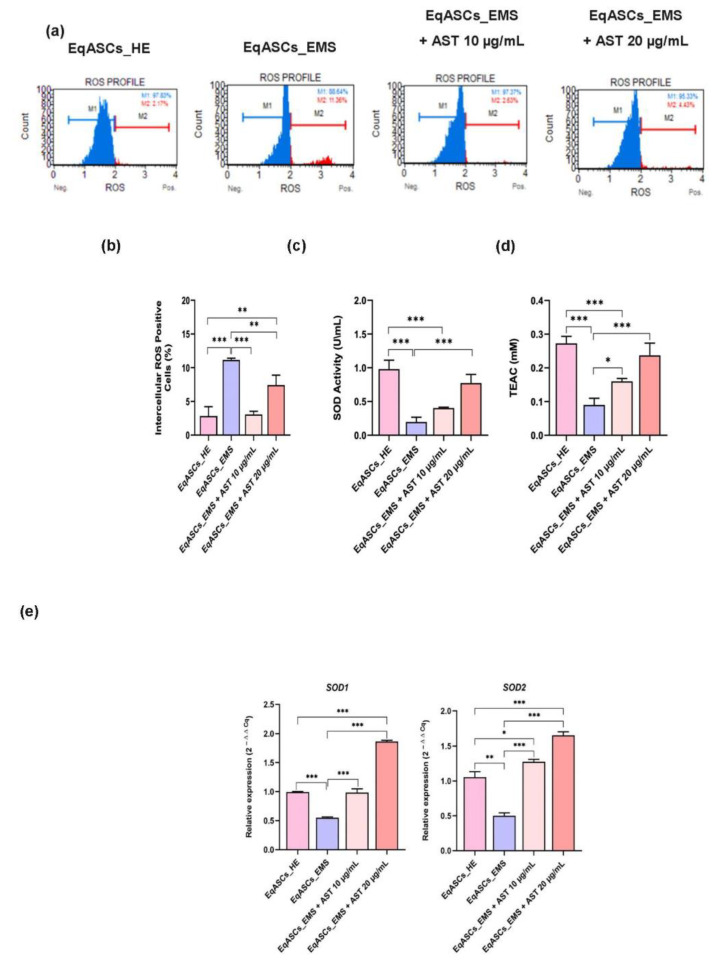
Effect of Astaxanthin on oxidative stress in EMS affected ASC cells. (**a**) Dot-Plots for intracellular ROS production detected by dihydroethidium (DHE) fluorescence staining. (**b**) Average percentages of total ROS+ cells in each experimental group. (**c**) Measurement of SOD Activity performed with Cayman Superoxidase Dismutase Assay Kit. (**d**) antioxidant capacity by Cayman’s Antioxidant Assay Kit. (**e**) Relative gene expression of *SOD1* and *SOD2* transcripts. A comparison of EMS, treatment groups and untreated healthy cells is shown by an asterisk (*). * *p* < 0.05, ** *p* < 0.01, *** *p* < 0.001.

**Figure 4 biomolecules-12-01039-f004:**
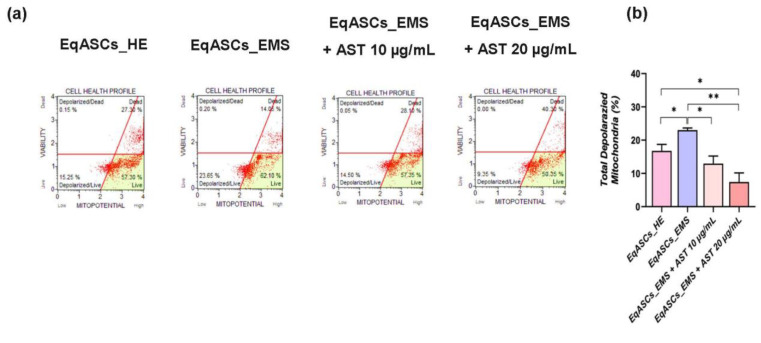
The effect of astaxanthin on mitochondrial dynamics in EMS ASC cells (**a**) Flow cytometric analysis of Mitochondrial membrane potential (MMP). (**b**) Percentages of total depolarized mitochondrial membrane potential. (**c**) Epi-fluorescent confocal microscope images of MitoRed stained cells; scale bar size 25 μm. (**d**) Imaris mitochondrial morphology analysis micrographs Mitochondrial morphology analysis. (**e**) Mitochondrial morphology analysis. (**f**) Representative Bar-Charts of the relative expression of mitochondrial fusion and mitophagy markers. (**g**) Levels of MFF, and Pink-1 were estimated with the western blot method. Relative expression was estimated using Image Lab software after normalization with β-actin (loading control). A comparison of EMS, treatment groups, and untreated healthy cells is shown by an asterisk (*). * *p* < 0.05, ** *p* < 0.01, *** *p* < 0.001. A hashtag (#) refers to a comparison of the EMS and astaxanthin treated groups. ### *p* < 0.001.

**Figure 5 biomolecules-12-01039-f005:**
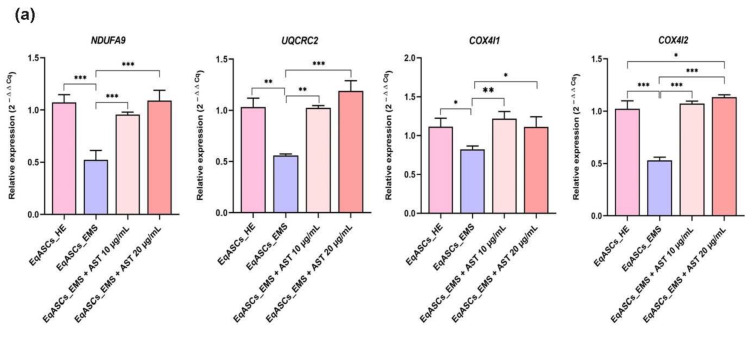
Effect of Astaxanthin on Mitochondrial metabolism. Expression of (**a**) genes encoding for the OXPHOS complexes (**b**) genes encoding of mitochondrial translational machinery regulators. A comparison of EMS, treatment groups and untreated healthy cells is shown by an asterisk (*). * *p* < 0.05, ** *p* < 0.01, *** *p* < 0.001.

**Table 1 biomolecules-12-01039-t001:** Gene expression.

Gene	Primer	Sequence 5′–3′	Amplicon Length (bp)	Accession No.
*Parkin*	F: R:	GTGCAGAGACCGTGGAGAAA GCTGCACTGTACCCTGAGTT	294	NM_013987.3
*Sod1 (Cu/Zn SOD)*	F: R:	CATTCCATCATTGGCCGCAC GAGCGATCCCAATCACACCA	130	NW_001867397.1
*Sod2 (Mn SOD)*	F: R:	GGACAAACCTGAGCCCCAAT TTGGACACCAGCCGATACAG	125	NW_001867408.1
*Pink1*	F: R:	GCTTGGGACCTCTCTTGGAT CGAAGCCATCTTGAACACAA	142	NM_032409.3
*Casp9*	F: R:	CAGGCCCCATATGATCGAGG CTGGCCTGTGTCCTCTAAGC	142	NM_032996.3
*Casp3*	F: R:	GGCAGACTTCCTGTATGCGT CCATGGCTACCTTGCGGTTA	167	XM_023630401.1
*Bcl-2*	F: R:	ATCGCCCTGTGGATGACTGAG CAGCCAGGAGAAATCAAACAGAGG	129	NM_000633.2
*p21*	F: R:	AGAAGAGGCTGGTGGCTATTT CCCGCCATTAGCGCATCAC	169	NM_001220777.1
*p53*	F: R:	AGATAGCGATGGTCTGGC TTGGGCAGTGCTCGCTTAGT	381	NM_001126118.1
*Casp8*	F: R:	ACTGTGATGTTGCTGGGACT CTTTCTCCTGGTGCATCTATCG	177	XM_001496753.4
*Bax*	F: R:	ACCAAGAAGCTGAGCGAGTGTC ACAAAGATGGTCACGGTCTGCC	356	XM_011527191.1
*Mfn1*	F: R:	GTTGCCGGGTGATAGTTGGA TGCCACCTTCATGTGTCTCC	146	NM_033540.3
*OPA1*	F:R:	CTTCTCTTGTTAGGTTCACCTGG TGTAAGAGAATGAGCTCACCAAG	110	XM_003363363.4
*GAPDH*	F: R:	GTCAGTGGTGGACCTGACCT CACCACCCTGTTGCTGTAGC	256	NM_001357943.2
*Wnt3*	F: R:	CACCTGCAAGTAGGGAGCCA GCTTCCCAGAGGACTTCGGT	80	XM_014739584.2
*NDUFA9*	F: R:	TTGGTATTCAGGCCACACCC GCTGGCTTCACGTCTTCAAC	103	XM_001494601.4
*UQCRC2*	F: R:	TGCTTCGTCTTGCATCCAGT AACTCCGGTGACGTGGTAAC	193	XM_001494381.5
*COX4I1*	F: R:	GAATAGGGGCACGAACGAGT GCCACCCACTCCTCTTCAAA	138	XM_023637444.1
*COX4I2*	F: R:	CCCCACCCCAGATGTTCT CGTGGTAGTTGGTGTAGGG	135	XM_005604417.3
*OXA1L*	F: R:	GACCTAGAAACCGTGGGACG GGAAGATCACTTGGCTCCCC	105	XM_008528958.1
*MRPL24*	F: R:	ATGATCCCTAGCGAAGCACC TGTAGAGACTCGTACCCGCT	123	XM_001500466.4
*MTERF4*	F: R:	CGCCACCTCCGTGCTATG CCCAAATGAGGGGCATCAGG	147	XM_023644068.1
*PUSL1*	F: R:	TCAGCCACTTCCAGGACCTA AGCCACATCCAAGCTGTCTG	120	XM_023636046.1
*TFAM*	F: R:	ATGATGGCTTTGAGTCCAGG CTAGATGATGGCGGGAGACTT	154	XM_023643450.1

Parkin Parkin RBR E3 ubiquitin protein ligase PARK2, Sod1 (Cu/Zn SOD) copper-zinc-dependant superoxide dismutase (CuZnSOD), Sod2 (Mn SOD) manganese-dependent superoxide dismutase (MnSOD), Pink1 PTENinduced putative kinase 1, Casp-9 caspase 9, Casp3 Caspase 3, Bcl-2 B cell lymphoma 2, p21 cyclin-dependent kinase inhibitor 1, p53 tumor suppressor p53, Casp-8 caspase 8, Bax BCl-2 associated X protein, Mfn-1 mitofusin 1, OPA-1 OPA1 Mitochondrial Dynamin Like GTPase, GAPDH glyceraldehyde 3-phosphate dehydrogenase, Wnt3 Wnt Family Member 3, NADH ubiquinone oxidoreductase subunit A9, UQCRC2 Ubiquinol-Cytochrome C Reductase Core Protein 2, COX4I1 Cytochrome c oxidase subunit 4 isoform 1, COX4I2 Cytochrome c oxidase subunit 4 isoform 2. MRPL24 Mitochondrial Ribosomal Protein L24, Mterf4 Mitochondrial Transcription Termination Factor 4, Tfam Mitochondrial transcription factor A, Pusl1 Pseudouridine Synthase Like 1, OXA1L mitochondrial inner membrane protein.

**Table 2 biomolecules-12-01039-t002:** List of antibodies employed for protein profiling using western blot analysis.

Antibody	Dilution	Catalog No.
PINK 1	1:1000	Biorbyt, orb331223
MFF	1:1000	Biorbyt, orb325479
β-Actin	1:1000	Sigma Aldrich, a2066

## Data Availability

Not applicable.

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
