# Peer review of "Astaxanthin Carotenoid Modulates Oxidative Stress in Adipose-Derived Stromal Cells Isolated from Equine Metabolic Syndrome Affected Horses by Targeting Mitochondrial Biogenesis"

_biomolecules, 2022, doi:10.3390/biom12081039_

Round 1

Reviewer 1 Report

In this manuscript the authors suggested that astaxanthin extract as has the potential to control and suppress oxidative stress, thereby improving the overall metabolic status of equine ASCs suffering from metabolic syndrome.

Despite being a rapidly developing area, in my opinion, this manuscript is superficial, because many of the indications and what is explained by the authors is not well-founded and for several topics the bibliographic references are not enough, particularly with regard to medical indications and medical properties. In the introduction, I suggested that the authors further elucidation the doses used, observed results and related mechanism in order to substantiate its use.

In the abstract, is it better change “extract” to “astaxanthin”?

Line 42: the excess space should be deleted. Check the entire document.

Line 48: there is no space before the reference, keep the format consistent, check the entire document.

Line 46-48: The sentence is not good expression, please revise?

Line 59, 120-127: “Several studies” should add references; “Many reports” should add references.

Line128-133: Write the highlights in meaningful way.

Line 139, 220-221: the excess space should be deleted between 3 and 200.

Line 140: a space should be added between numbers and units “5g”, except “℃”, check the entire document.

Line 272: capitalize the first letter.

Table: use three-wire grid.

Results: add discussion or related reports in other literature.

Reviewer 2 Report

I have had the honor of reviewing the manuscript entitled "Astaxanthin carotenoid modulates oxidative stress in adipose-derived stromal cells isolated from equine metabolic syndrome affected horses by targeting mitochondrial biogenesis", the work presents interesting novel information on possible treatments for equine metabolic syndrome.

The author’s work presents a novel dietary treatment approach for the equine metabolic syndrome.

Although the work presented is of adequate quality, I would still have to point some minor suggestions for revision.

The introduction is missing many references, for instance in page 1 line 40 "which affects both humans and animals", in the text comprehended between lines 57 - 60, and many other claims in the introduction section. This must be revised, and appropriate references added.

The methods selected are adequate for hypothesis validation. Methods are well described.

The results are well presented and concise.

The discussion section discusses the results of the experiment and compares the obtained results with some human experiments however the discussion lacks direct comparisons with other equine metabolic syndrome models, and possibly other models related. While the discussion of intracellular pathways is too extent, more on the practical applicability of the information should be discussed. References are also missing in the discussion section as is the case for lines 641 – 642 “initially, astaxanthin has an antioxidant effect, eliminating reactive oxygen and nitrogen species and other free radicals.” As well as for other sentences in the text.

Conclusion are in relation with the results.

Round 2

Reviewer 1 Report

The manuscript has been well revised.